# The Understanding and Compact Modeling of Reliability in Modern Metal–Oxide–Semiconductor Field-Effect Transistors: From Single-Mode to Mixed-Mode Mechanisms

**DOI:** 10.3390/mi15010127

**Published:** 2024-01-12

**Authors:** Zixuan Sun, Sihao Chen, Lining Zhang, Ru Huang, Runsheng Wang

**Affiliations:** 1School of Integrated Circuits, Peking University, Beijing 100871, China; 2School of Electronic and Computer Engineering, Peking University Shenzhen Graduate School, Shenzhen 518055, China

**Keywords:** MOSFET, mixed-mode reliability, hot carrier degradation (HCD), bias temperature instability (BTI), self-heating, off-state degradation (OSD), time-dependent dielectric breakdown (TDDB), electromigration (EM)

## Abstract

With the technological scaling of metal–oxide–semiconductor field-effect transistors (MOSFETs) and the scarcity of circuit design margins, the characteristics of device reliability have garnered widespread attention. Traditional single-mode reliability mechanisms and modeling are less sufficient to meet the demands of resilient circuit designs. Mixed-mode reliability mechanisms and modeling have become a focal point of future designs for reliability. This paper reviews the mechanisms and compact aging models of mixed-mode reliability. The mechanism and modeling method of mixed-mode reliability are discussed, including hot carrier degradation (HCD) with self-heating effect, mixed-mode aging of HCD and Bias Temperature Instability (BTI), off-state degradation (OSD), on-state time-dependent dielectric breakdown (TDDB), and metal electromigration (EM). The impact of alternating HCD-BTI stress conditions is also discussed. The results indicate that single-mode reliability analysis is insufficient for predicting the lifetime of advanced technology and circuits and provides guidance for future mixed-mode reliability analysis and modeling.

## 1. Introduction

To sustain the scaling down of complementary metal–oxide–semiconductors (CMOS), new materials and structures have been continuously incorporated into the design and manufacturing of CMOS devices, such as High-κ/metal gate (HKMG) [1,2,3], strain technology [4,5,6], and multi-gate transistors [7,8,9,10]. For the past 50 years, CMOS devices have evolved from planar transistors with micron-level channel lengths to gate-all-around (GAA) transistors with sub-twenty nanometer channel lengths [11,12,13,14,15,16]. However, with the scaling down of CMOS, the continuous application of new technologies, and the increasing complexity of manufacturing processes, reliability issues are gradually emerging as a significant challenge in device applications [17,18,19,20,21,22,23]. 

In practical circuit operations, devices experience various reliability issues, triggering non-ideal effects such as circuit functional aging and failure. As shown in Figure 1, taking a typical inverter circuit as an example, devices undergo three typical biasing conditions: gate voltage (V_gs_) > 0 V, drain voltage (V_ds_) = 0 V; |V_gs_| > 0 V, |V_ds_| > 0 V and V_gs_ = 0 V, |V_ds_| > 0 V. The degradation phenomena observed in NMOS or PMOS devices under the bias condition of |V_gs_| > 0 V, V_ds_ = 0 V are, respectively, termed positive bias temperature instability (PBTI) [24,25,26] and negative bias temperature instability (NBTI) [27,28,29,30,31,32,33]. The degradation phenomenon observed when the device is under the bias condition of |V_gs_| > 0 V, |V_ds_| > 0 V is referred to as hot carrier degradation (HCD) [34,35,36,37,38,39,40,41]. The degradation observed under the bias condition of V_gs_ = 0 V, |V_ds_| > 0 V is termed off-state degradation (OSD) [42,43,44]. Meanwhile, during device operation, the devices also face failure issues such as time-dependent dielectric breakdown (TDDB) [45,46,47] and electromigration (EM) [48,49,50].

In previous research, in-depth studies have been conducted on the degradation mechanisms and aging models of individual degradation modes. For instance, the accepted mechanism for BTI is the capture/emission of carriers by oxide traps, leading to a significant degradation in the threshold voltage [51,52]. The hydrogen bridges and hydroxyl-E’ are widely acknowledged as the origins of BTI traps [53,54,55]. For HCD, the phenomenon involves the breakage of Si-H bonds by high-energy hot carriers, forming interface states that induce degradation in the threshold voltage and mobility [56,57,58]. As device nodes advance and carrier energy decreases, electron–electron scattering (EES) and multiple vibration excitation (MVE) mechanisms have been proposed to explain the contributions of low-energy carriers to HCD [59,60,61]. Simultaneously, the contribution of oxide trap-induced degradation in HCD becomes more pronounced, especially in FinFET devices, where HCD is considered a combined effect of oxide traps and interface states [62,63]. In comparison to HCD and BTI, there is less research on OSD due to its less pronounced occurrence in long-channel devices. OSD encompasses the combined action of carriers and secondary carriers generated by impact ionization, resulting in a reduction in threshold voltage or non-monotonic changes [64]. Based on the corresponding mechanisms of different degradations, compact models are proposed for predicting aging. In the past, research on the failure mechanisms of TDDB and EM mainly focused on gate dielectric breakdown under V_g_ stress and source/drain metal electromigration [65,66,67,68] by establishing a failure extrapolation model to constrain circuit design, such as maximum voltage and minimum metal line width.

However, in practical device and circuit applications, devices do not undergo single-mode reliability issues; instead, they operate under mixed-mode issues [69,70,71]. Previous research has predominantly focused on pure single-mode reliability and independently modeled each mechanism. This approach, which is based on the prediction of aging using a single-mode reliability mechanism, fails to accurately predict the lifetimes of advanced devices. Especially with the increasingly scarce circuit design margin today, complex circuit systems design based on advanced devices demand more precise aging prediction models to ensure the most appropriate design margins [72,73]. Therefore, recent research has gradually turned its attention to the study of the mechanisms and models of mixed-mode reliability. 

In this paper, we provide a comprehensive review of the mechanisms and the model concerning mixed-mode reliability. The mixed-mode reliability mechanisms and the impact of alternating stress were discussed separately. First, we summarized the mixed-mode aging mechanisms, including the hot carrier degradation with self-heating effect, hot carrier degradation with inhomogeneous BTI, and OSD, and provided existing modeling methods. We also analyzed the mixed-mode failure mechanisms of on-state TDDB and gate metal electromigration. Then, we summarized the HCD-BTI alternating stress degradation. It is underscored that employing a single-mode reliability model for modeling proves inadequate in accurately predicting mixed-stress aging in advanced devices. Consequently, there arises a critical imperative to establish a prediction model and framework based on mixed-mode reliability. 

## 2. Mixed-Mode Reliability Mechanisms

### 2.1. Hot Carrier Degradation with Self-Heating Effect

The introduction of materials with low thermal conductivity and 3D device geometries results in FinFET devices having limited heat dissipation capability and more severe self-heating effects [74,75,76,77]. Previous studies on the self-heating effect of FinFET technology have shown a significant impact on performance and HCD [78,79,80,81]. As shown in Figure 2, the self-heating dominant region coincides with the HCD stress region. Thus, the impact of the self-heating effect on HCD requires a thorough investigation.

To investigate the hot carrier degradation with a self-heating effect, two key issues need to be addressed: (1) how to accurately characterize the self-heating effect of the device and (2) how to analyze the temperature dependence of hot carrier degradation accurately. There are several methods for characterizing the self-heating effect, such as the Pulse IV method [82,83], the gate resistance method [84], the heat-sensor method [85], etc. The Pulse IV method involves comparing the current under low-duty-cycle ultra-narrow pulse signals with DC I-V to extract the self-heating effect. This method relies on an ultra-fast measurement platform to ensure that the pulse width is less than the device’s thermal time constant (τ < 10 ns) and provides enough scattering time to avoid thermal accumulation effects. The gate resistance method requires a device with a specific gate structure, applying voltage to both ends of the four-terminal gate and testing the gate current at the other two ends, inferring the channel temperature by comparing the change in gate resistance. Similarly, the heat-sensor method relies on a common source configuration, where one MOS device acts as the “heat” and the adjacent MOS acts as the “sensor”. When voltage is applied to the “heat” device, its self-heating effect raises the ambient temperature around the “sensor” device, affecting the transfer characteristics of the “sensor” device. However, the temperature around the “sensor” device will be lower than the actual self-heating temperature of the “heat” device, leading to an underestimation of the self-heating effect.

After accurately characterizing the self-heating effect of FinFETs, it is essential to incorporate the temperature influence into the degradation of hot carriers. Therefore, the temperature dependence of HCD becomes a crucial aspect to explore further. As mentioned earlier, the degradation amount of HCD in long-channel devices decreases with increasing channel temperature, showing negative temperature dependence due to the single vibration excitation (SVE) mechanism mainly dominating in long-channel devices, as elevated temperature increases lattice scattering, reducing carrier energy [86,87]. Conversely, in short-channel devices, the degradation amount of HCD increases with rising channel temperature, exhibiting positive temperature dependence due to the MVE mechanism mainly dominating in short-channel devices, as higher temperature enhances carrier energy [88,89]. However, recent studies have found that HCD in advanced process node devices exhibits non-uniform temperature dependence, meaning that the temperature dependence changes with different bias conditions and stress times [90,91], as shown in Figure 3.

The cause of non-uniform temperature dependence in HCD is the varying activation energy (E_a_) of different traps [37,90]. The proportions of various traps also change with different bias conditions, ambient temperatures, and degradation times, leading to a macroscopic change in the E_a_ of hot carrier degradation. Viewing this non-uniform temperature dependence from the perspective of trap generation mechanisms reveals that it is caused by the change in the dominant mechanism of trap generation under different stress conditions. 

The coupled degradation of hot carriers and self-heating effects can be reflected in the frequency dependence and layout dependence of HCD. As shown in Figure 4, with the increase in the device frequency, self-heating effects gradually decrease, leading to a reduction in HCD degradation [92]. Meanwhile, multi-fin devices experience more severe self-heating effects due to mutual heating, resulting in more severe HCD for multi-fin devices [93], as shown in Figure 5. However, these studies are based on characterization results under worst-case stress conditions (V_gs_ = V_ds_). In actual circuits, devices are not always or continuously under worst-case stress conditions. Due to the non-uniform temperature dependence of HCD, the degradation of HCD does not necessarily increase with the increase in self-heating effects in the full bias map. Thus, hot carrier degradation may have different layout and frequency dependencies in different bias ranges.

### 2.2. Hot Carrier Degradation with Inhomogeneous Bias Temperature Instability

Generally, under bias conditions where V_gs_ > 0 V and V_ds_ > 0 V, the predominant degradation is commonly attributed to hot carrier degradation. However, across a wide range of {V_ds_, V_gs_} bias maps, the degradation is not solely governed by a single mechanism but rather exhibits a mixed-mode degradation mechanism, as shown in Figure 6. In previous studies, the mixed HCD and inhomogeneous BTI degradation existed in V_gs_ > 0 V, V_ds_ > 0 V stress conditions, as shown in Figure 7 [94]. Hence, the pressing challenge is to analyze the contributions of inhomogeneous BTI components and pure HCD components and establish an accurate predictive model. 

One approach involves analyzing the contribution of inhomogeneous BTI from the perspective of the vertical electric field distribution in the channel, thereby isolating the contribution of pure HCD [95]. It has been observed that under HCD stress, BTI exhibits inhomogeneous distribution. This is due to changes in channel potential and carrier distribution influenced by non-equilibrium transport. Early attempts involved establishing a simple model for inhomogeneous BTI degradation by calculating the impact of V_ds_ on channel potential. For example, at V_gs_ = V_ds_, the inhomogeneous BTI degradation is approximately 50% of that under BTI stress alone, as shown in Figure 8. Based on this approach, a consolidated model for HCD degradation decoupling analysis and inhomogeneous BTI contributions can be established [39]. The degradation contribution ratio of pure HCD and inhomogeneous BTI under different frequency {|V_gs_| > 0, |V_ds_| > 0} stresses can be decoupled and analyzed across the entire voltage domain, as shown in Figure 9. When the frequency reaches 1 MHz, the contribution of fast traps is neglected, leading to a reduced proportion of BTI components. However, at 1 GHz, the self-heating effect of the device decreases. For nFinFET, the oxide trap 2 of HCD has a larger E_a_ than the PBTI trap, causing an increased proportion of PBTI at GHz. In contrast, for pFinFET, the NBTI trap has a larger E_a_, resulting in a reduced proportion of NBTI at GHz. This analytical approach can effectively estimate inhomogeneous BTI degradation but overlooks the impact of secondary effects in non-equilibrium transport. Subsequent research revealed that secondary carriers generated by non-equilibrium transport affect the carrier distribution near the source region, influencing inhomogeneous BTI degradation. This study suggests that high V_ds_ bias not only reduces the occupancy probability of traps at the drain region but also affects traps near the source region [94,96].

### 2.3. Mechanisms in Off-State Degradation

The off-state bias condition (|V_gs_| = 0 V, |V_ds_| > 0 V) is a common bias condition in practical circuit operations. Therefore, studying the aging mechanisms and modeling of devices under off-sate conditions is crucial for predicting device lifetime and designing aging-aware circuits. In HKMG planar devices, more publications report that off-state stress leads to an increase in on-state current degradation and a decrease in threshold voltage, also called the hot-electron-induced punch-through (HEIP) effect [97,98,99], as shown in Figure 10. This degradation phenomenon is explained as secondary carriers being captured by traps near the drain region, causing a decrease in effective channel length, resulting in reduced threshold voltage and an increase in leakage current. However, in advanced FinFETs, off-state degradation is considered a complex phenomenon involving multiple electrical traps and mechanisms [100,101]. As shown in Figure 11, the non-monotonic shift of threshold voltage caused by the contribution of multiple electrical traps has been observed in FinFETs.

For PMOS, secondary electrons caused by band-to-band tunneling and impact ionization (I/I) are observed in the channel-drain region. These electrons are trapped by the oxide field into gate oxide traps near the drain region, such as PBTI. The high-energy part is accelerated by the lateral electric field, breaking Si-H bonds near the source region, like electron-induced HCD (eHCD), resulting in a significant decrease in |V_th_| and degradation of mobility. Simultaneously, high-energy hole injections from the source region under lateral electric field acceleration can also break Si-H bonds near the drain region, causing Vth degradation similar to hole-induced HCD (hHCD). Furthermore, both eHCD and hHCD lead to degradation in subthreshold swing and mobility. Similar mechanisms exist in NMOS. For NMOS, secondary holes are trapped by the electric field into gate oxide traps near the drain region, such as NBTI, or break Si-H bonds in the source region under lateral electric field acceleration, like hHCD. Simultaneously, the high-energy part of electrons can also break Si-H bonds in the drain region, causing V_th_ degradation similar to eHCD. Since HCD is almost irreversible, PBTI in pFinFET (NBTI in nFinFET) contributes to the recoverable part of off-state stress degradation. It is worth noting that the main HCD mechanisms are different for different device types. Due to the higher injection energy of holes (4.7 eV) and the lower saturation value of interface states, hHCD in nFinFET can be neglected, but in pMOS, where holes are the main charge carriers, hHCD cannot be ignored. Therefore, in nFinFET, the main trap types are NBTI and eHCD traps, while in pFinFET, the main traps are PBTI, hHCD, and eHCD traps. Based on the above mechanism, HCD and BTI models are employed to describe the degradation contributions of each component in OSD. The BTI recovery model is utilized to predict OSD recovery, as the recoverable traps of BTI contribute to the recovery of OSD. This model can effectively predict both the degradation and recovery of the threshold voltage. Simultaneously, compact models for subthreshold swing and mobility degradation are established based on the saturated power-law model due to the depletion of the available Si–H bond.

### 2.4. On-State Time-Dependent Dielectric Breakdown with Self-Heating Effect

In planar devices, much research has focused on conventional gate-only time-dependent dielectric breakdown (V_gs_-only TDDB) under stress conditions where the gate voltage is applied without drain voltage (V_gs_ > 0, V_ds_ = 0). On the other hand, on-state TDDB (V_gs_ > 0, V_ds_ > 0) has received less attention due to the reduction in the gate oxide field caused by applying drain voltage, leading to improved lifetime for planar devices under on-state TDDB conditions. However, with the advancement to FinFET technology at advanced technology nodes, the impact of on-state TDDB on device lifetime becomes more severe, as shown in Figure 12 [102,103]. Experimental results also indicate that the on-state TDDB lifetime of FinFET decreases with increasing drain bias. A widely accepted explanation is that the self-heating effect under on-state conditions facilitates TDDB. With a larger drain bias, the device has higher power dissipation, and the more severe self-heating effect accelerates the breakdown of the gate oxide. TEM characterization analysis has revealed that under on-state TDDB stress conditions, breakdown occurs near the middle of the Fin, as shown in Figure 13. The observations show that under on-state TDDB stress, the “hillock” on the silicon Fin penetrates through the dielectric layer (the dielectric breakdown-induced epitaxy (DBIE)) [102]. In previous reports, the DBIE phenomenon of V_gs_-only TDDB occurred at the bottom of Fin [104]. The breakdown point has shifted from the bottom of the Fin in V_gs_-only TDDB to the middle of the Fin in on-state TDDB. Simulations using multi-physics field simulation, considering self-heating effects and electric field distribution, have illustrated that a local hot spot in the middle of the Fin is critical to the occurrence of breakdown in the middle. Thus, the lifetime prediction model for on-state TDDB not only needs to consider the extrapolation of the gate electric field but also the influence of dissipated power [103]. 

However, prior research did not consider the influence of hot carriers in on-state TDDB, so understanding the role of hot carriers in on-state TDDB and how to characterize their impact remains an area for further exploration. 

### 2.5. Gate Metal Electromigration with On-State Soft Breakdown

Due to the extremely weak gate current, gate metal electromigration is typically overlooked in device and circuit design. However, past research has revealed that a mixed mode of self-heating effects and soft breakdown can lead to gate metal electromigration [97]. As shown in Figure 14, TEM images illustrate non-uniform contrast in the M0 layer after a soft breakdown in the on-state, indicating gate metal electromigration. Simultaneously, as soft breakdown deteriorates, gate leakage current increases, aggravating gate metal electromigration and resulting in void formation. The results indicate tungsten metal ions diffuse to the gate through Via. Other metal ions, such as titanium from Via filler, remain uncontaminated. Correspondingly, there is no occurrence of gate metal electromigration under V_gs_-only soft breakdown stress conditions. Simulations considering gate leakage current and self-heating effects indicate that the combined effect of increased leakage current and self-heating is a significant factor causing gate metal electromigration. A layout design featuring a double via is proposed to significantly mitigate gate metal electromigration failure by reducing gate metal current and self-heating effects.

It is worth noting that as gate metal lines and transistors are tightly scaled at advanced nodes, gate metal electromigration reliability becomes more severe. However, addressing this issue is crucial for enhancing the future reliability design of advanced circuits.

## 3. Impact of Alternating Stress Conditions

During the operation of circuits, devices typically undergo complex alternating stress conditions. In digital circuits, devices are often influenced by BTI and off-state stress during signal stability, while during signal transitions, they are affected by HCD. In addition, some analog circuits (such as ADCs, amplifiers, etc.) may operate under alternating HCD and BTI stresses. Therefore, in practical circuit aging analysis, analyzing aging under alternating bias conditions becomes crucial. Past research indicates that models simply superimposing single degradation mechanisms struggle to precisely match real experimental data, as shown in Figure 15 [95,105,106,107]. This is attributed to the influence of two major factors: (1) The HCD phase includes contributions from inhomogeneous BTI. When analyzing alternating HCD-BTI stresses, it is necessary to consider the historical effect of inhomogeneous BTI. (2) The influence of HCD stress on the recovery process of BTI traps. The secondary carriers from HCD can significantly enhance the recovery effect of BTI reported in the previous study, as shown in Figure 16. This may be attributed to the substantial generation of secondary electrons in PMOS under severe V_ds_ stress, leading to the excessive emission of NBTI traps [108]. Therefore, in establishing a mixed degradation model, it is necessary to consider the inhomogeneous BTI degradation within HCD. Simultaneously, it is necessary to consider the impact of the historical effects of inhomogeneous BTI on the pure BTI stage and the influence of the historical effects of pure BTI on inhomogeneous BTI. In the recovery stage, it is essential to develop BTI recovery models under different V_ds_ voltages to accurately describe the recovery of BTI during the HCD stress stage.

Moreover, for FinFETs, the self-heating effect also plays a crucial role in mixed mode degradation under alternating stress conditions. The transient self-heating effect introduced during the HCD stage will have an impact on the degradation and recovery stages of BTI in alternating HCD-BTI stress conditions. Thus, in the actual circuit simulation process, the HCD-BTI mixed stress model with a transient self-heating effect needs to be established [71]. As shown in Figure 17, a mixed-mode HCD-BTI aging prediction framework containing transient self-heating effects is proposed. This framework has been implemented and validated using silicon data. The heating and cooling stages are divided into multiple time intervals, incorporating the historical effects of BTI degradation at different temperatures through the introduction of the effective time (t_eff_) concept. Therefore, when assessing the degradation of complex alternating stress waveforms, one not only needs to consider multiple mechanisms for constructing aging models but also faces the challenge of the low computational efficiency of complex mixed aging models.

Consequently, numerous frameworks integrating machine learning for aging evaluation have been introduced, aiming to enhance prediction efficiency while ensuring accuracy and minimizing invasiveness. Recurrent neural networks (RNNs) are a widely used neural network architecture. Their distinctive recurrent concept and, most importantly, the structure of long short-term memory (LSTM) networks enable them to perform well in addressing and predicting sequential data problems. Considering the compatibility of the continuous-time equations of RNNs with transient circuit simulations, RNNs demonstrate applicability in modeling aging circuits [109]. While in practical applications of circuit simulation, an RNN model, with its internal feedback, may not be stable in circuit simulation. In [110], the support vector machine (SVM) model is used to capture the relationship between signal probabilities and delay degradation of cells. However, the impractical assumptions about constant supply voltages and temperatures cause an obvious loss of accuracy. A versatile aging-aware delay model for generic cell libraries has been introduced, utilizing transistor-level SPICE simulations and feed-forward neural networks (FFNNs), which demonstrates that the proposed model achieves fast estimation of the aging-induced delay with high accuracy close to transistor-level simulation [111]. Machine learning (ML) methods can be used to predict the aging delay in transistors and cell circuits, which map the device degradation to the aging delay of cell circuits, thereby substantially amplifying prediction efficiency [112,113,114,115]. Despite the promise of potentially replacing aging simulations, there is still a need to integrate both fresh and stress simulations into the aging evaluation process, especially in emerging usage scenarios. Of particular significance is the observation that their input features lack considerations for circuit structure, resulting in a realized enhancement in efficiency that falls short of attaining optimal levels. In [116], an innovative modeling framework is introduced for rapid aging-aware timing analysis, utilizing a temporal-spatial graph neural network (GNN). This framework employs a gated tanh unit (GTU) as the temporal network, extracting device degradation from dynamic biases. Simultaneously, it incorporates inductive GraphSAGE as the spatial network to gather comprehensive graph information from circuit topology and output circuit aging delay. This pioneering approach, distinguished by its exceptional feature capture capability, markedly enhances prediction efficiency, particularly within the context of nano-scale technology.

In summary, these reported works collectively contribute to advancing the understanding and modeling of aging effects in modern circuits, offering diverse solutions to the challenges posed by predicting circuit aging under complex waveforms from neural network-assisted to temporal-spatial GNN approaches.

## 4. Conclusions

In this paper, we have reviewed recent research on the mixed-mode reliability of MOSFET. The introduction of advanced devices has led to more complex aging and failure mechanisms due to the self-heating effects. This complexity exacerbates the difficulty in establishing accurate compact models. Moreover, a more accurate and reliable compact model can only be established through research that better aligns with the mixed-mode reliability encountered in practical device usage. Especially in the current scenario where design margins are becoming increasingly constrained, taking a crucial step from studying single-mode reliability to investigating mixed-mode reliability is essential. This shift is necessary to develop more precise lifetime prediction models that can support large-scale advanced circuit designs. However, with further advancements in device technology and the introduction of overly complex mechanism couplings, coupled with new materials and structures, analyzing the mechanisms behind mixed-mode reliability will face significant challenges. It prompts us to contemplate whether AI technology holds promise for future mixed-mode reliability analysis and modeling. It is undeniable that AI has its limitations in establishing accurate reliability prediction models. For instance, how neural networks can train correct compact model formulas when the underlying mechanisms are unclear poses a significant challenge. Different compact model formulas may exhibit excellent fitting accuracy for short-term degradation, but significant variations in extrapolated lifetimes are inevitable. Therefore, addressing the challenge of establishing accurate lifetime prediction models for increasingly complex mixed-mode reliability in the future remains an imminent task.

## Figures and Tables

**Figure 1 micromachines-15-00127-f001:**
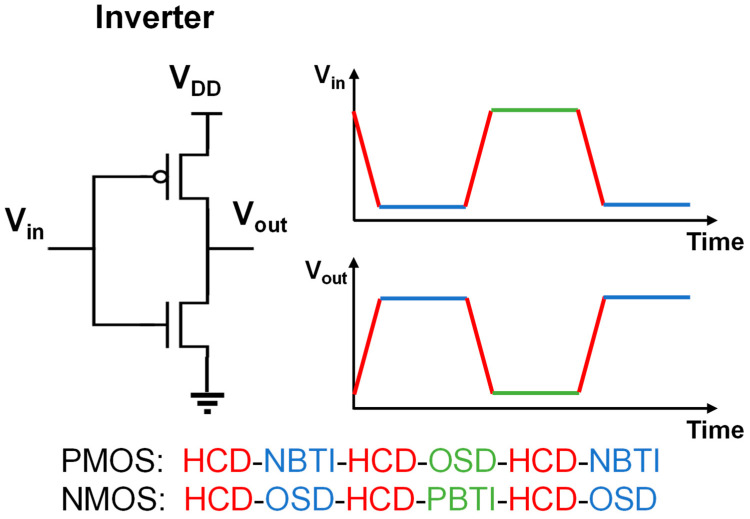
Schematic of the different degradation modes in an inverter.

**Figure 2 micromachines-15-00127-f002:**
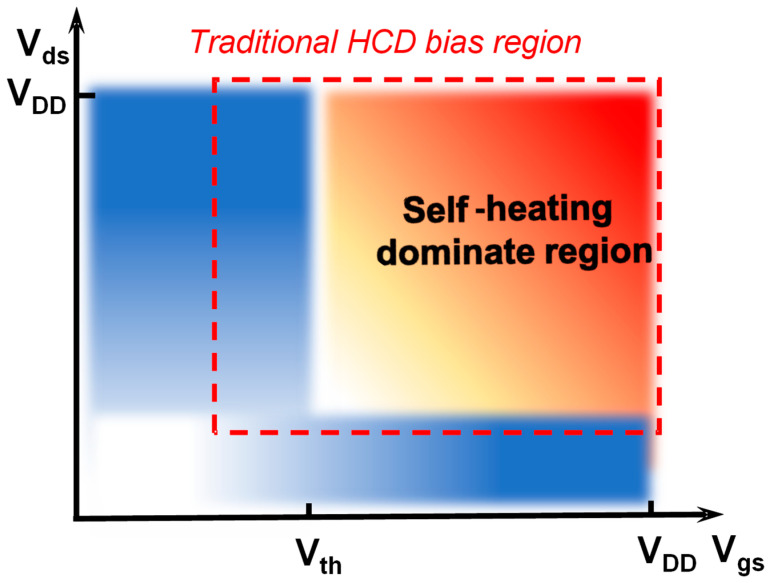
Schematic of the self-heating dominate region in the full {V_gs_, V_ds_} bias map. The self-heating dominant region overlaps with the HCD region.

**Figure 3 micromachines-15-00127-f003:**
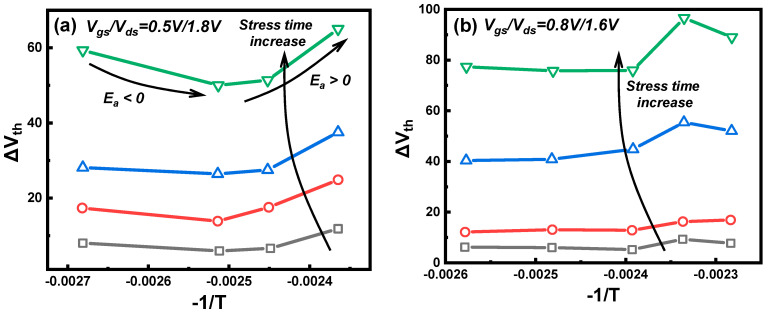
Experimental results of nFinFET show the non-universal HCD temperature dependence. (**a**) V_gs_ < V_ds_ bias condition; (**b**) V_gs_ > V_ds_ bias condition. Data from Ref. [90].

**Figure 4 micromachines-15-00127-f004:**
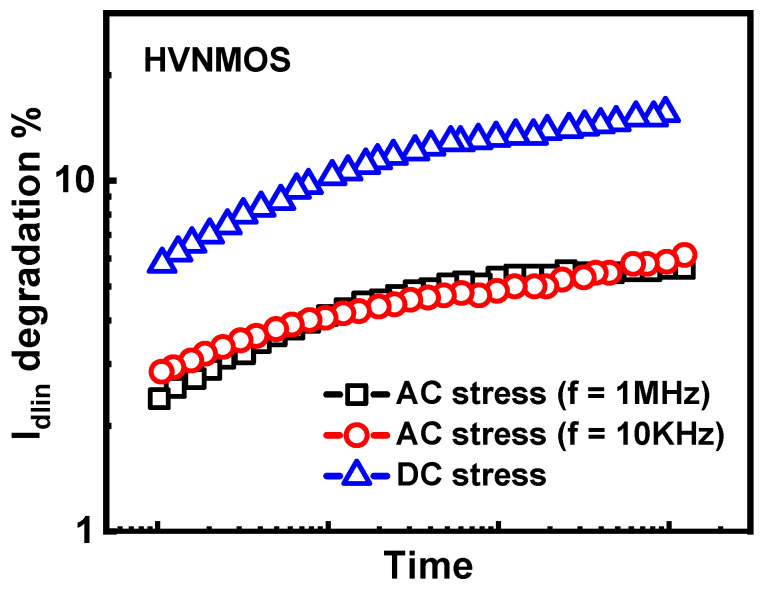
Frequency dependence of HCD in HVNMOS; HCD decreases as the frequency (duty cycle = 0.5) increases. Data from Ref. [92].

**Figure 5 micromachines-15-00127-f005:**
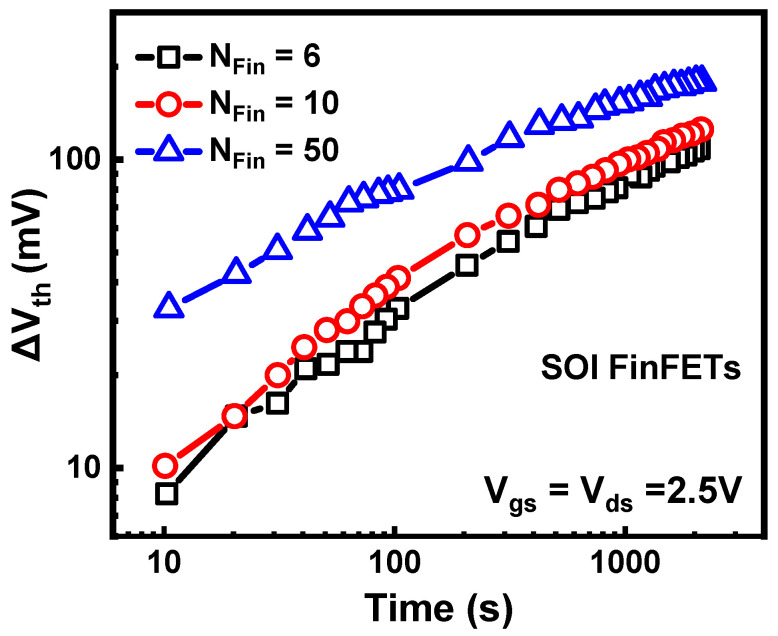
Layout dependence of HCD in SOI FinFETs; HCD becomes increasingly severe as the Fin number increases under stress condition of V_ds_ = V_gs_ = 2.5 V. Data from Ref. [93].

**Figure 6 micromachines-15-00127-f006:**
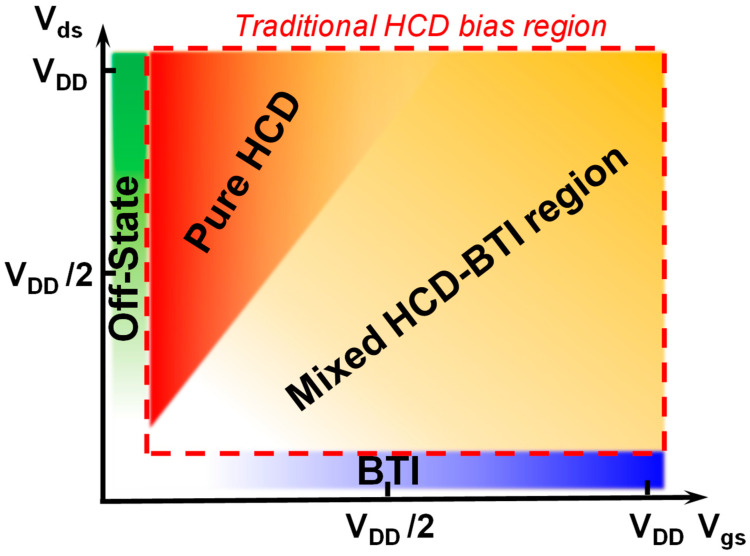
Schematic of the dominant aging region in the full {V_gs_, V_ds_} bias map. Devices suffer mixed HCD-BTI degradation in V_gs_ > 0 V, V_ds_ > 0 V bias condition region.

**Figure 7 micromachines-15-00127-f007:**
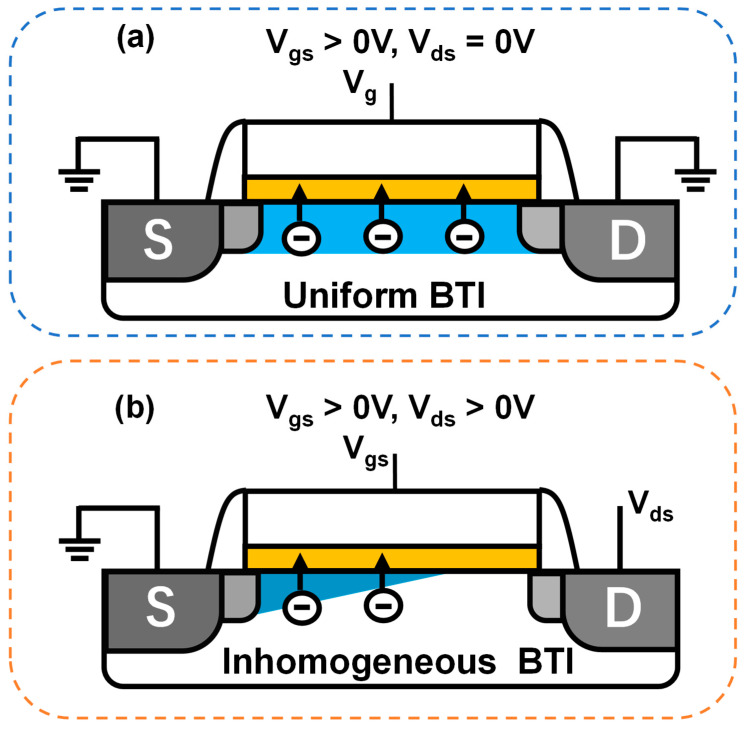
Schematic of the uniform BTI vs. inhomogeneous BTI. (**a**) Device suffers unifrom BTI under V_gs_ > 0 V bias condition. (**b**) Device suffers inhomogeneous BTI under V_gs_ > 0 V, V_ds_ > 0 V bias condition.

**Figure 8 micromachines-15-00127-f008:**
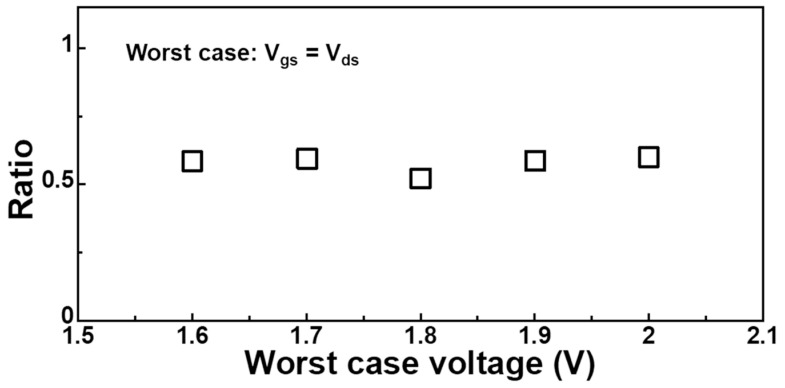
The ratio of inhomogeneous BTI and uniform BTI under V_gs_ = V_ds_ condition. Data from Ref. [95].

**Figure 9 micromachines-15-00127-f009:**
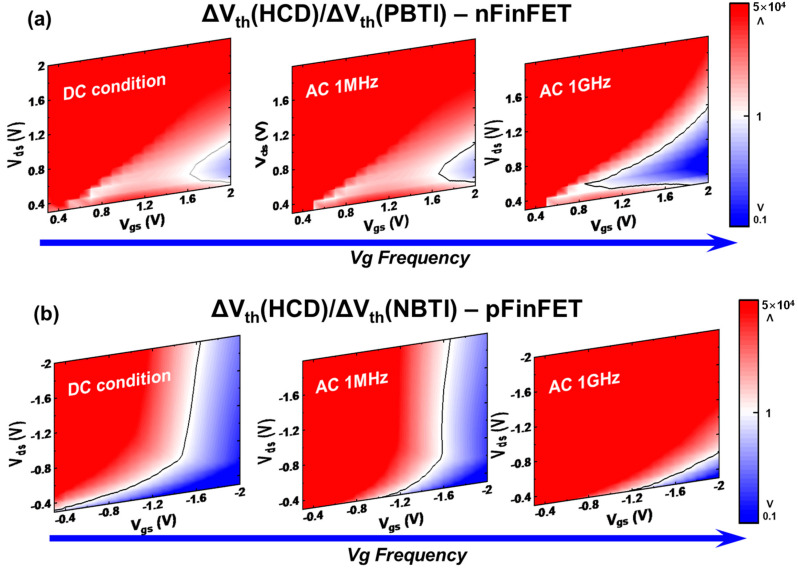
The ratio of HCD and BTI contributions in full map with self-heating effect under different V_gs_ stress frequencies (V_gs_ is AC or DC signal with duty factor DF = 0.5, V_ds_ is DC bias): (**a**) nFinFET; (**b**) pFinFET.

**Figure 10 micromachines-15-00127-f010:**
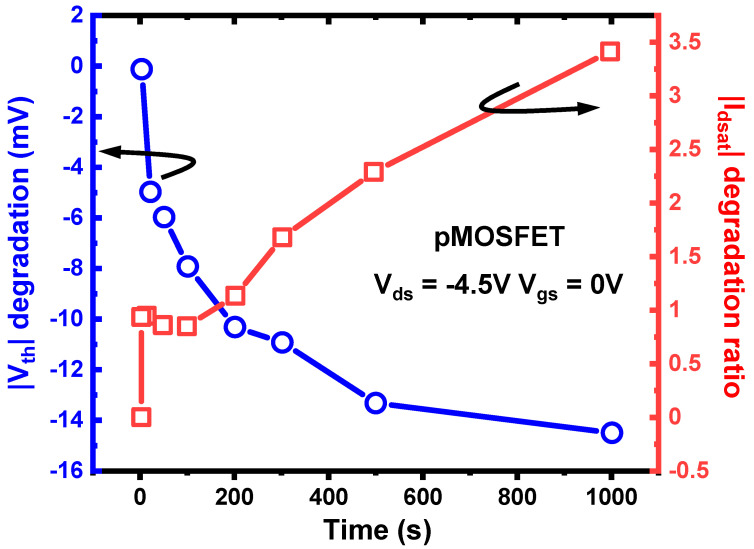
Experimental results of OSD in pMOSFET—V_th_ decrease and I_dsat_ increase after OSD. Data from Ref. [97].

**Figure 11 micromachines-15-00127-f011:**
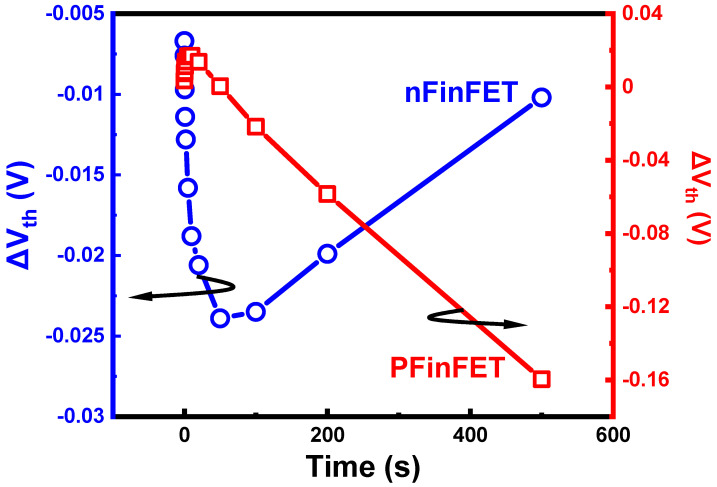
Experimental results of OSD in FinFET: the non-monotonic shift of threshold voltage caused by the contribution of multiple electrical traps. Data from Ref. [101].

**Figure 12 micromachines-15-00127-f012:**
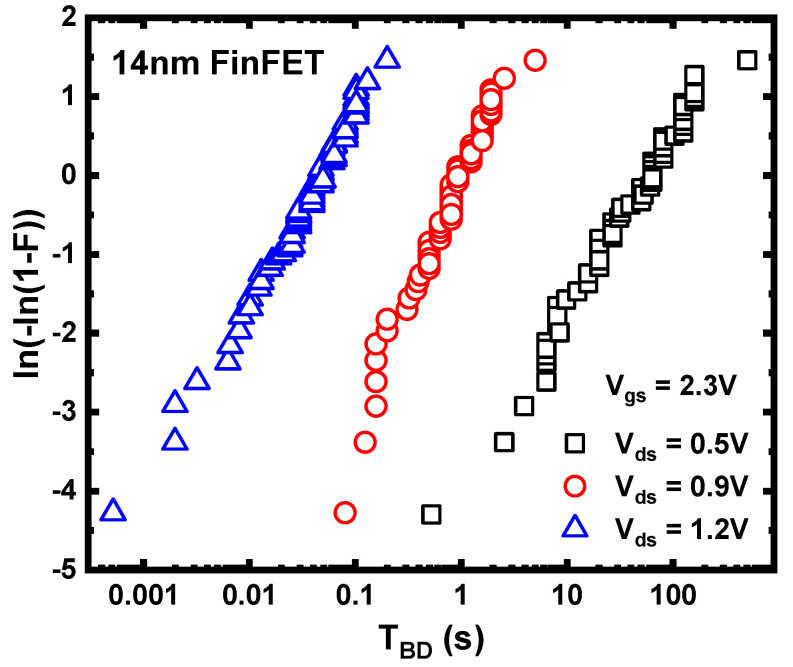
Weibull slops of on-state TDDB with different V_ds_ bias. As the V_ds_ increases, the lifetime of TDDB decreases. Data from [103].

**Figure 13 micromachines-15-00127-f013:**
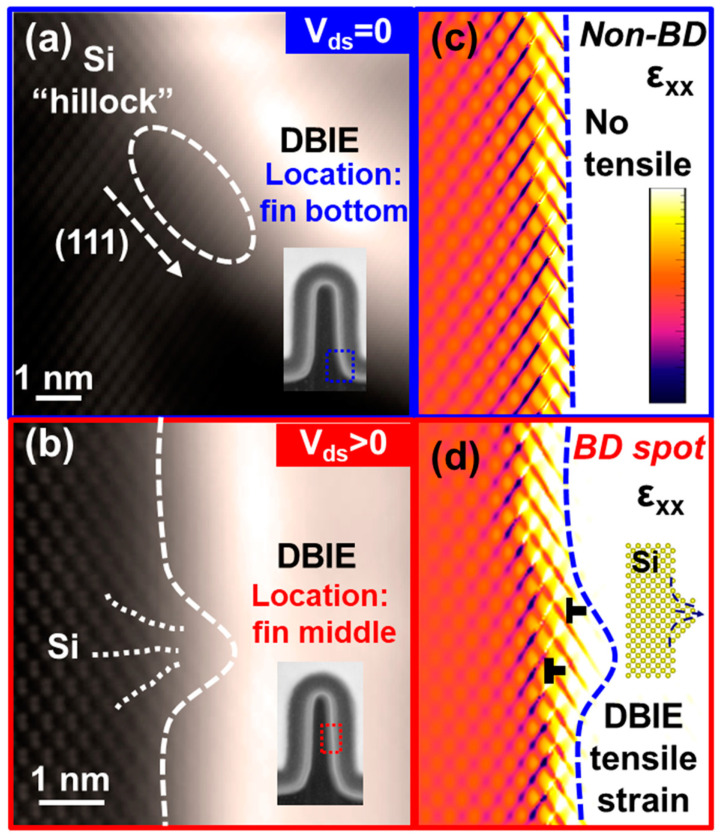
The comparison of high-resolution transmission electron microscope images of dielectric breakdown-induced epitaxy morphologies compared in (**a**) V_gs_-only TDDB and (**b**) on-state TDDB; comparison of tensile strain analysis in (**c**) V_gs_-only TDDB and (**d**) on-state TDDB. Tensile strain variations induced by DBIE near the Fin middle have been observed after on-state TDDB [102]. Copyright (2023) The Japan Society of Applied Physics.

**Figure 14 micromachines-15-00127-f014:**
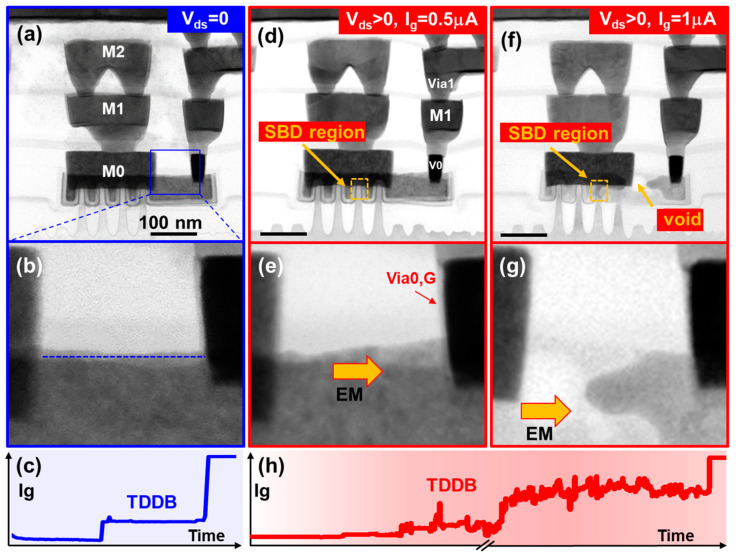
Gate metal electromigration evolution from V_gs_-only TDDB to on-state TDDB. (**a**–**c**) No gate metal electromigration after V_gs_-only TDDB; (**d**,**e**) gate metal electromigration occurred after on-state soft breakdown; (**f**–**h**) gate metal electromigration intensifies with the increase of gate current in on-state TDDB [102]. Copyright (2023) The Japan Society of Applied Physics.

**Figure 15 micromachines-15-00127-f015:**
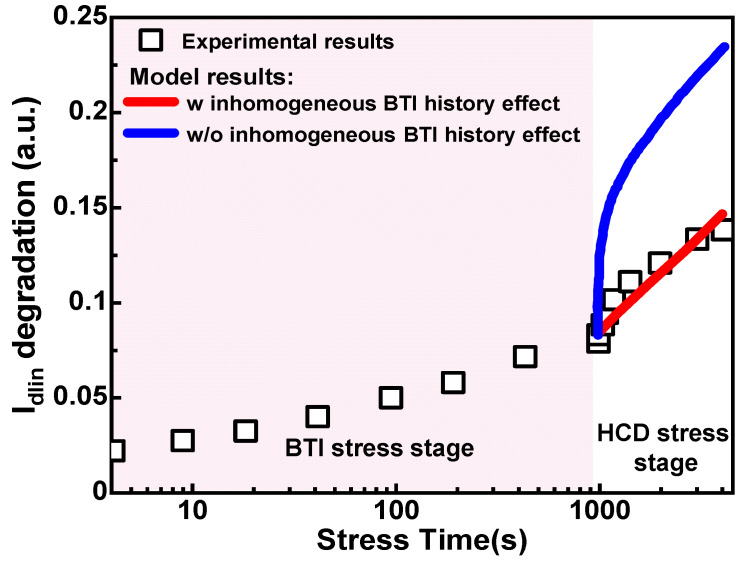
The comparison of experimental data and model prediction. The model with simple superposition of a single degradation mechanism cannot predict the experimental data and needs to consider the historical effect of inhomogeneous BTI during the HCD stage. Data from Ref. [105].

**Figure 16 micromachines-15-00127-f016:**
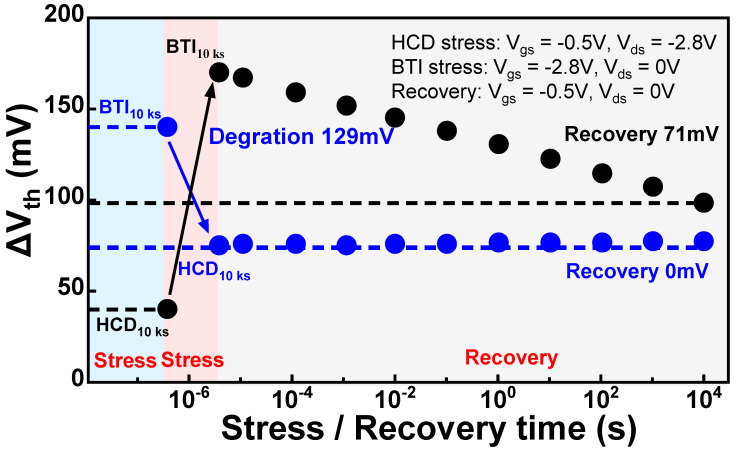
Measurement results of alternating HCD-BTI stress conditions. BTI recovery exhibits an abnormal recovery trend starting from the HCD stress stage. Data from Ref. [108].

**Figure 17 micromachines-15-00127-f017:**
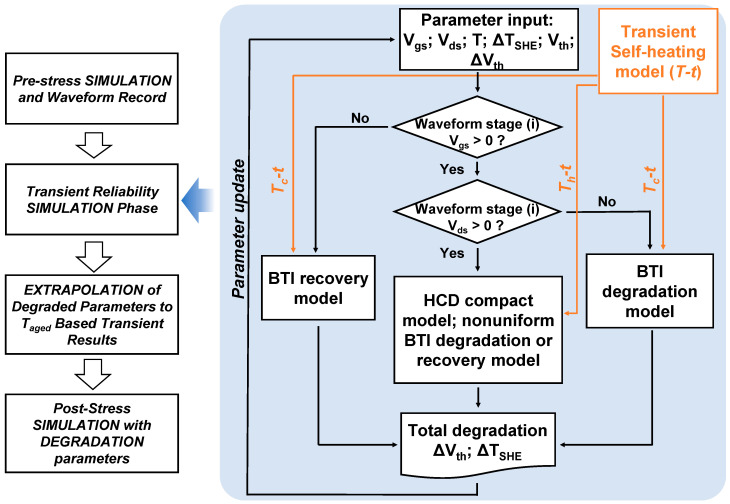
The mixed-mode stress simulation flow with transient self-heating effects [71]. Firstly, input device characteristics and bias waveforms. If V_gs_ > 0 V and V_ds_ > 0 V, consider the impact of the transient heating stage on HCD, inhomogeneous BTI degradation, and BTI recovery effects. If V_gs_ > 0 V and V_ds_ = 0 V, consider the residual high temperature during the cooling process and its impact on BTI degradation. If V_gs_ = 0 V and V_ds_ = 0 V, consider the residual high temperature during the cooling process and its impact on BTI recovery. Copyright (2023) IEEE.

## Data Availability

Data are contained within the article.

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
