# Peer review of "The Understanding and Compact Modeling of Reliability in Modern Metal–Oxide–Semiconductor Field-Effect Transistors: From Single-Mode to Mixed-Mode Mechanisms"

_micromachines, 2024, doi:10.3390/mi15010127_

Round 1
Reviewer 1 Report
Comments and Suggestions for Authors
The article provides an overview of the mechanisms and compact models of mixed-mode reliability, discussing the mechanisms and modeling methods.
Issues related to integrated circuit reliability are quite complex. While the intention of this paper is commendable, the overall quality needs improvement.
The narrative contains numerous transitions, and the expression of viewpoints is not sufficiently clear, making it easy for readers to become confused. The quality of figures and tables in the article needs enhancement, for example, Figures 2 and 6 lack both quantitative and accurate qualitative information. Additionally, there are errors in the captions of Figures 4 and 5, where the titles are upside down. Figures 11, 12, 13, and 14 are all sourced from reference [97], which does not align with the diversity expected in the references of a review article. Moreover, the figure captions in other charts lack sufficient descriptions of the images. The article introduces various failure mechanisms and failure analyses, but there is limited discussion on the modeling of hybrid mode reliability. The focus of modeling lies both in analysis and prediction, and further research is needed in this regard.
Given the significant issues in the article, it is recommended to enhance the quality before resubmission for further review.
Comments on the Quality of English LanguageModerate editing of English language required.
Reviewer 2 Report
Comments and Suggestions for Authors
In the paper, the author reviewed recent research on the mixed-mode reliability of MOSFET. This review is comprehensive and readable. The manuscript could be accepted in its present form.
Comments on the Quality of English LanguageThe quality of English is good.
Author Response
Thank you very much for the careful reading and the positive comments on our manuscript. We have addressed some minor English errors in revised manuscript.
Reviewer 3 Report
Comments and Suggestions for Authors
This article is an extremely comprehensive review of aging mechanisms for current state-of-the-art MOSFET devices. The information presented within is useful for members of the community at all levels of expertise, especially for those looking to acquaint themselves with the current issues surrounding aging/degradation mechanisms in FET devices. The article is well-written and comprehensive in its current form, however I believe that it can be presented in an even clearer format. These suggestions are mostly aimed towards making this clearer for the younger audience, as I see this review as excellently poised to be a "go-to" resource for many young researchers interested in entering this field.
I found this manuscript to be scientifically-sound in its analysis and compilation of results and I believe Micromachines to be a good match for publication.
1.) Vds, Vgs are not properly defined until lines 180-182. For experts in the field, source and gate voltage notations are well-understood. However, for ease of access I suggest putting the definition of these voltages somewhere towards the start of the paper, particularly before line 38 where typical biasing conditions are noted.
2.) SHE is never explicitly defined before it is used in line 93. By reading the text it is clear that it is "self-heating effect", but explicitly defining this will help with clarity. Similarly, SBD (line 213) and HKMG (line 226) are also not defined in the text.
3.) The sentence "...cause of non-uniform temperature dependence in HCD is the varying activation energy..." needs a citation.
4.) The labels for Figure 4 and 5 seem to be swapped. Either in the text or in the Figure caption.
5.) It was unclear if the new mixed-mode framework was at all implemented? Was this new framework at all tested?
6.) AI and ML techniques were not brought up until halfway through the concluding paragraph. Is there a precedent already set to using these techniques? For example, if other groups have begun to think about this idea as well, it would be useful to bring up this idea earlier in the article.
Round 2
Reviewer 1 Report
Comments and Suggestions for Authors
The manuscript has been improved in quality and is more comprehensive after the revisions. I recommend acceptance for publication. However, please note that there are still issues with the numbering of figures in the revised manuscript. Kindly cross-check them carefully in conjunction with the main text.